# A CMOS-compatible oscillation-based VO$_2$ Ising machine solver

Olivier Maher [1,2] ✉, Manuel Jiménez [3], Corentin Delacour [4], Nele Harnack[1], Juan Núñez[3], María J. Avedillo[3], Bernabé Linares-Barranco [3], Aida Todri-Sanial[4,5], Giacomo Indiveri [2] & Siegfried Karg[1] ✉

Phase-encoded oscillating neural networks offer compelling advantages over metal-oxide-semiconductor-based technology for tackling complex optimization problems, with promising potential for ultralow power consumption and exceptionally rapid computational performance. In this work, we investigate the ability of these networks to solve optimization problems belonging to the nondeterministic polynomial time complexity class using nanoscale vanadium-dioxide-based oscillators integrated onto a Silicon platform. Specifically, we demonstrate how the dynamic behavior of coupled vanadium dioxide devices can effectively solve combinatorial optimization problems, including Graph Coloring, Max-cut, and Max-3SAT problems. The electrical mappings of these problems are derived from the equivalent Ising Hamiltonian formulation to design circuits with up to nine crossbar vanadium dioxide oscillators. Using sub-harmonic injection locking techniques, we binarize the solution space provided by the oscillators and demonstrate that graphs with high connection density ($\eta > 0.4$) converge more easily towards the optimal solution due to the small spectral radius of the problem's equivalent adjacency matrix. Our findings indicate that these systems achieve stability within 25 oscillation cycles and exhibit power efficiency and potential for scaling that surpasses available commercial options and other technologies under study. These results pave the way for accelerated parallel computing enabled by large-scale networks of interconnected oscillators.

Combinatorial optimization problems (COPs) find deep roots in several industrial applications, such as drug synthesis, resource allocation, computer vision powered by artificial intelligence, and circuit layout design[1]. Their relevance extends beyond computational theory, finding widespread applicability across various industries and remaining one of the most prevalent challenge faced by computer scientists[2]. A multitude of COPs belong to a class of problems solvable in nondeterministic polynomial time (NP)[3]. Finding the solution to NP-complete problems using computing technologies based on the traditional von Neumann architecture introduces a range of challenges related to latency, convoluted interconnections, and are difficult to integrate in compact electronic systems, considering that energy consumption scales exponentially with the complexity of the problem[4].

This limitation in performance has generated a growing need for accelerated computing with new types of algorithms and chip

[1]IBM Research Europe - Zurich, Säumerstrasse 4, 8803 Rüschlikon, Zürich, Switzerland. [2]Institute of Neuroinformatics, University of Zürich and ETH Zürich, Winterthurerstrasse 190, 8057 Zürich, Switzerland. [3]Instituto de Microelectrónica de Sevilla, IMSE-CNM (CSIC, Universidad de Sevilla), Av. Américo Vespucio 28, 41092 Sevilla, Spain. [4]LIRMM, University of Montpellier, 56227 Montpellier, France. [5]Eindhoven University of Technology, Electrical Engineering Department, 5612AZ Eindhoven, Netherlands. ✉e-mail: OGM@zurich.ibm.com; SFK@zurich.ibm.com

designs[5]. In these designs, memory and computation are embedded together and work in synergy on the same medium to create new neuromorphic architectures inspired by nature[6,7]. Consequently, novel computing paradigms are being explored to replicate some of the brain's fundamental operations to solve combinatorial optimization problems more efficiently[4,7–11]. In these efforts, researchers have focused on harnessing the potential of new phase-change materials, which exhibit high performance and unprecedented power efficiency in the analog domain[2,12,13]. The goal is to combine multiple inputs linearly and nonlinearly to process information, thereby implementing a rich catalog of operations similar to those performed by neurons[14]. In this work, we study a potential architecture based on coupled oscillators that offers adaptability, error tolerance, and flexibility to achieve the level of complexity observed in modern computers, while operating with much lower energy consumption and latency figures[15]. We choose to work with vanadium dioxide ($VO_2$), which distinguishes itself from other phase-change materials by displaying an intrinsic structural phase transition at a temperature (68 °C) that can be rapidly triggered by heating from room temperature[16,17]. Additionally, $VO_2$ offers essential features for the fabrication of coupled oscillator-based neural networks (ONNs), such as: **1**. Scalability, **2**. Low-power and high-frequency operation, **3**. Robustness against noise, **4**. Easily interfaceable with high-fanout electronic interconnections, **5**. High endurance, and **6**. CMOS compatibility[18]. Unlike other van der Pol oscillator-based technologies, our $VO_2$-based oscillators do not introduce complex levels of nonlinearity governed by higher order differential equations, making them easier to couple and capable of tackling large-scale problems represented by large interconnected networks[2]. $VO_2$-based oscillating neural networks are dynamical systems both complex enough to encode computationally heavy problems within the heuristic domain and simple enough to realize with simple connections ensuring stable problem-solving and solution convergence under small programming biases[2]. Oscillators made of $VO_2$ have been shown to exhibit extremely high energy efficiency, operating orders of magnitude lower than digital CMOS oscillators[19]. Foreseen scalability and energy efficiency play an essential role in our device; operating at levels far below the threshold voltages of transistors, where leakage currents have become the main limiting factor for further computing performance improvements[20].

These new $VO_2$ oscillators would not completely supplant CMOS technology but rather complement it, particularly in areas where step-by-step instruction-based computation is insufficient[2]. Our integrated devices make a stride towards analog computers, leveraging their inherent parallel processing capabilities encoded in the physical state of individual units[1]. This approach has the potential to outperform digital computers that rely on parallelization across multiple processors by employing the Ising model to build specialized purposed machines for solving NP-Complete problems[1,19,21,22]. This model allows for the representation of any problem in the NP complexity class as an Ising problem, with a polynomial time cost instead of an exponential one[1]. By using this formulation, it is possible to carry information in the phase and/or the frequency ($f$) of a signal to achieve richer data mapping and enhanced robustness to voltage-noise scaling issues[18,23]. This is why ONNs have gained popularity in solving optimization problems, employing various technologies such as bulky LC oscillators, ring oscillators with latch-based coupling, quantum and photonics equivalents, discrete-time memristors, and CMOS-based oscillators[19,24]. However, the successful physical implementations of these technologies are quite limited and mainly motivated by simulation-based results[2].

In our work, we advance towards realizing capacitively coupled oscillators for solving COPs, specifically Graph Coloring, Max-cut, and Maximum 3-Satisfiability (Max-3SAT) problems[8–10,17,25,26]. We experimentally investigate the benefits of injecting harmonics into the system to help discretize outputs, a finding we previously reported[27,28]

through simulation, and successfully couple up to 9 $VO_2$-based oscillators. Furthermore, we analyze the solvability of problems with fewer interconnections or minor variability between individual oscillators, providing insights into their limitations and potential solutions.

## Results
### $VO_2$ oscillators
Individual units emulating neurons are built to realize self-sustained oscillations from a direct-current (DC) energy source. The hysteretic nature of the phase transition in $VO_2$, depicted in Fig. 1a, creates a region of instability in the current-voltage (I-V) curve when connected in series with a resistive device[8,10,18,19]. By carefully selecting an appropriate resistance value, such that the load line intersects the unstable region defined by the transition points, the system can follow the same electrical trajectory illustrated in Fig. 1b, d, switching periodically from a high (insulator) to a low (metallic) resistive state[29,30]. The introduction of this external series resistance generates a dynamic voltage divider, establishing the relaxation oscillation nature of the system with fixed limit cycle, amplitude, and frequency[18,19,30]. The addition of an external load capacitor connected off-chip, as shown in Fig. 2, is employed to achieve uniform frequency operation among the oscillators and compensate for variations in parasitic capacitances resulting from individual contacts to our $VO_2$ devices (see Fig. 1c)[31]. This configuration prevents refractory period effects observed in other studies[32] and results in stable oscillations, with less than 3.5% amplitude variation from cycle to cycle (see Fig. 1d)[33].

The $VO_2$ devices are fabricated on silicon platform with a hafnium oxide interlayer to create a granular film comprised between two metallic electrodes whose cross-section define an area where current can flow and generate thermal filaments through Joule heating (see Fig. 1c)[31,34]. Our manufacturing process follows semiconductor industry standards, facilitating the integration of our devices at the Back-

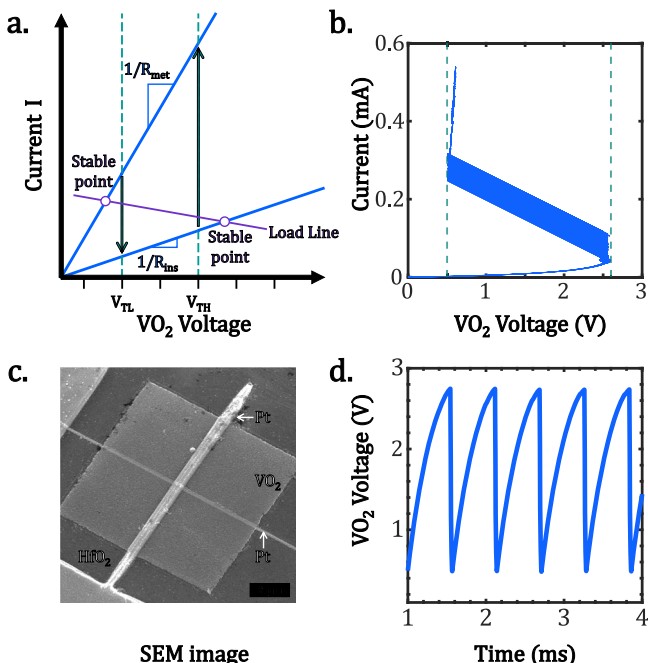

**Fig. 1 | $VO_2$ devices characteristics. a** I-V characteristics schematic of a $VO_2$ device showing phase-transition points at low ($V_{TL}$) and high ($V_{TH}$) threshold voltages. The intersection of the load line creates stable points outside the hysteresis window, resulting in self-sustained oscillations. **b** I-V measurements on a **c** crossbar $VO_2$ device (active area: $100 \times 50 \times 60$ nm$^3$) connected to a series resistance ($R_s$) biased with a voltage ramp ranging from 0 V to 5 V. **d** Oscillation measurements of a single $VO_2$ crossbar oscillator at 220 K (active area: $80 \times 80 \times 60$ nm$^3$, $R_s = 50$ kΩ, $C_L = 11.27$ nF, $V_{DD} = 5$ V).

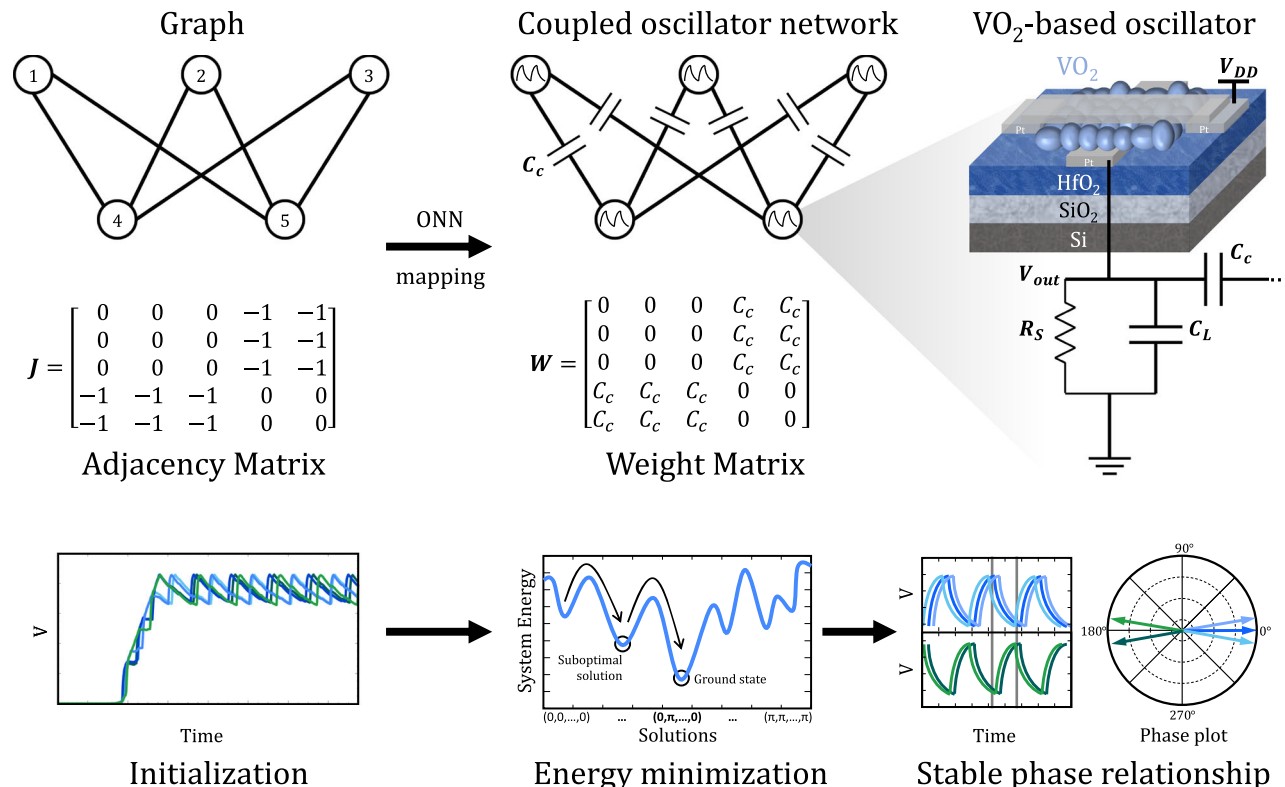

**Fig. 2 | Schematic representation of the mapping realized between an optimization problem's graph and an ONN.** Nodes become VO$_2$ oscillators and vertices, coupling capacitors (C$_c$). Each oscillator consists in one crossbar VO$_2$ device connected to a series resistance (R$_s$) and a load capacitor (C$_L$). The computational dynamics of the system evolve from an initial point to a stable phase relationship in the search for the lowest energy state.

End-of-Line (BEOL) while ensuring their compatibility with CMOS technology. The detailed fabrication process and the basics on fundamental operation can be found in Maher et al.[33] and in Supplementary Information (SI). The typical oscillating behavior of one device is shown in Fig. 1d. Low temperature operation is further motivated in SI.

## Dynamics of coupled VO$_2$ oscillators
A VO$_2$-based ONN consists of a system of oscillators acting as neurons, interconnected with synaptic weights, representing the coupling strength and the memory of the network[9]. The interaction between coupled oscillators has been extensively investigated and can be described by the Kuramoto model[18,23]. The Kuramoto formalism analytically derives the phase dynamics in a network of interconnected oscillators, demonstrating their ability to synchronize and lock in frequency[18,23,30,35]. Although the model is restricted to the "weak-coupling" regime and assumes nearly identical oscillators, it still applies to a wide range of physical oscillation-based systems and computing tasks[12,18]. In real physical systems, the ease with which VO$_2$ oscillators can be connected becomes the prevailing criterion to successfully realize computing[18,35]. While hybrid systems with local plasticity and high bandwidth capabilities can be designed using programmable resistive devices, capacitive coupling is the preferred method for connecting oscillators to solve NP-complete problems[6,30]. The high-pass filtering properties of this configuration guarantee synchronization without mutually altering the devices' DC operating points[30].

In the case of electrical oscillators, such as our VO$_2$-based relaxation oscillators, the off-chip implementation of passive and capacitive interconnection provides direct and sufficiently strong interaction to bring the oscillators into frequency locking[2,31,35]. This is due to the dynamic exchange of non-dissipative power between the oscillators that ensures synchronization[30]. When a coupling capacitance is introduced between only two oscillators, it leads to a modified frequency through the resulting effective capacitance, as demonstrated by Parihar et al.[30], while the phases tend to stabilize in the out-of-phase configuration, 0 and π respectively, if the coupling value is at least approximately 1% of the net capacitance of each oscillator[19]. When oscillators are interconnected with several others like in Fig. 2, the resulting phase of each oscillator is the combined effect of the repelling forces originating from each connection[17]. One can make the most out of this property to effectively map and solve fundamental optimization problems, prioritizing efficiency over optimality.

## Graph Coloring problem
The Graph Coloring problem consists in assigning a color to the nodes in a graph using the minimum number of colors, such that no connected nodes share the same color[17,36].

In Fig. 3, the experimental results of four graph coloring problems involving three to six oscillators are shown. The input geographical problem is mapped onto a network of VO$_2$ oscillators, where the coupling capacitors represent borders between individual countries. The circuit rapidly converges to a steady state within 10 oscillation cycles, a significantly faster process compared to testing all potential combinations. The stability of the phase relationships can be noted in the time-dependent phase plots found in Fig. 3, illustrating the progression of phases relative to an initial state (center of the plot) to their ultimate state (exterior of the plot) in reference to a designated device. Table S3 (SI) provides details regarding the different parameter values used in each experimental set, along with corresponding waveforms showcasing the stable state in Fig. 3. The parameter sets should be considered as possible examples. Other combinations where the supply voltage (V$_{DD}$) and the series resistance (R$_s$) are adjusted with the devices' active area could work as well. An adaption of the parameters for graphs with different number of nodes and edges is required as the capacitances vary. This makes it evident that there is a considerable

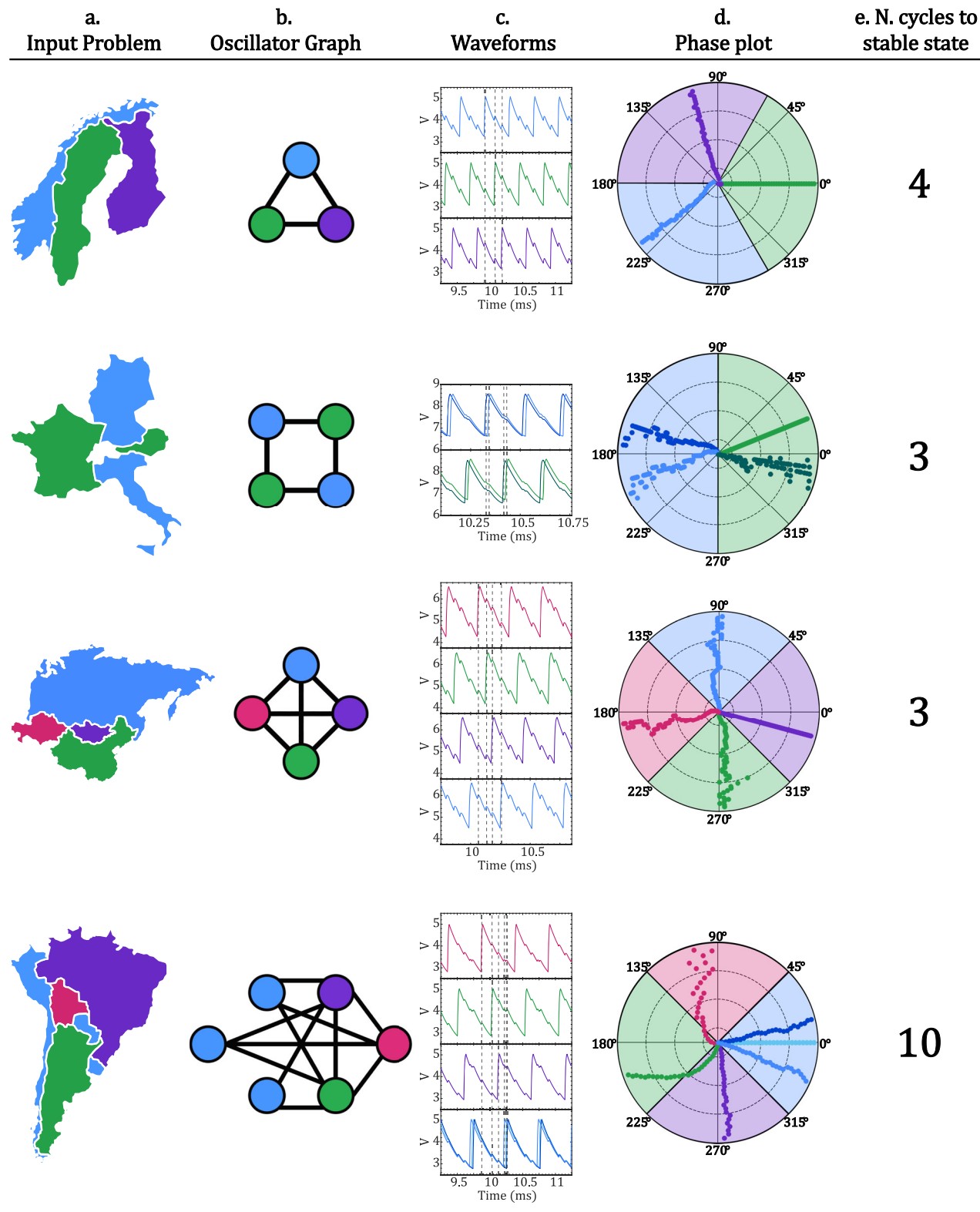

**Fig. 3 | Experimental results of coupled relaxation VO₂ oscillators, involving 3 to 6 nodes to solve the Graph coloring problem. a** Input problem, **b** Oscillator graph, measured **c** Waveforms and **d** Phase relationships of the ONN outputs. **e** Number of cycles required to get to the stable state. For the most complex graph, the system converges to the solution within 10 oscillation cycles, wherein the phase order defines a color assignment for each node.

challenge in establishing any form of metric for selecting values to color our map and converge rapidly towards a solution[17]. In our case, the careful choice of these values resulted in a cluster diameter, which represents the maximum phase difference among oscillators sharing

the same color grouping, that is relatively small[17]. For the Central European and South American graphs, the cluster diameter averaged 33.5° and 30.0°, respectively. The inherent sparsity of these graphs without all-to-all connectivity makes coloring more challenging[17],

resulting in larger cluster diameters compared to the Northern Europe and East Asia graphs. In the South America graph, characterized by nonuniform connectivity with varying degrees of connections on each node, the combined and unbalanced repelling effect of the coupling capacitances establishes a phase ordering among the oscillators[17]. This phase ordering can only approximate the minimum vertex coloring, causing uneven cluster spacing in the solution[17,37]. More effective mapping techniques employing a circular ordering in color assignment could be employed to further minimize the cluster diameter, particularly for larger-scale graphs[17].

### Ising formulation of problems

Having demonstrated the ability of our VO₂-based ONN to solve optimization tasks such as Graph Coloring, we now explore more complex COPs where variables (oscillators) are constrained to binary states (spins) within their Boolean expressions. We focus on two of Karp's 21 NP-complete problems: Max-cut and 3-SAT[3,38–40]. Mapping these problems into an Ising Hamiltonian equation can be achieved by setting the appropriate coupling coefficients in the energy function[39,41]:

$$H = -\sum_{1 \le i \le j \le n} J_{ij}s_i s_j - \sum_{i=1}^{n} \boldsymbol{h}_i s_i \qquad (1)$$

Where $J_{ij}$ is a coupling coefficient between units $i$ and $j$, which can be positive or negative and is usually achieved in an ONN using resistors[10] or capacitors[26], respectively.

$s_i$ is the spin (up ↑ or down ↓) of unit $i$, which can be mapped in a binarized network of oscillators by the phase (0 or π).

And $\boldsymbol{h}_i$ is an external bias, representing the interaction of unit $i$ with an external unit with fixed spin value.

Mohseni et al.[1] and Wang et al.[39] demonstrated that by minimizing the system's energy expressed by the Hamiltonian, the system's state naturally evolves towards the ground state, effectively solving the COP. Analyzing the system's energy while considering its oscillatory nature through the Kuramoto model under specific conditions shows that the ONN minimizes a Lyapunov equation[23,39,41]. When the states are simplified to a binary form, this equation is equivalent to the Ising Hamiltonian, ensuring the system evolves to a lower and stable energy state[39,41]. The intrinsic physical phase evolution of the coupled oscillators towards this attractor state, as shown in Fig. 2, is exploited to solve the Hamiltonian equation empirically as a complete and reliable Ising machine[19].

### Max-cut problem

The Max-cut problem consists in partitioning a weighted graph into a binarized state of two sub-graphs where the weights of the edges between them are maximized[3,40] – see Fig. 4. In this problem, specific conditions include dropping the second term in the energy function ($\boldsymbol{h}_i = 0$ in Eq. (1)). Due to capacitive coupling connecting multiple oscillators, the phases of neighboring oscillators tend to diverge towards polar opposites[41]. This cumulative effect of mutual interactions renders it impossible to maintain a binary state (0 and π)[41,42]. One strategy to deal with this challenge involves the introduction of a Sub-Harmonic Injection Locking (SHIL) signal[18,27,39,43]. When this signal lies sufficiently close to a multiplying factor $\boldsymbol{N}$ of the natural frequency of the oscillators ($f_{osc}$), i.e. $f_{SHIL} = \boldsymbol{N} \times f_{osc}$, phase synchronization occurs to force the phases of all the units within one of the $\boldsymbol{N}$ phase groups that are exactly $2\pi/\boldsymbol{N}$ apart[26]. We fix $\boldsymbol{N}$ to 2 to ensure binarization for the Max-cut problem. Solving this problem can be done quite efficiently by combining the effect of noise with subharmonic injection. The resulting fluctuations in energy enhance the probability that the system escapes from local minima (non-optimal solutions) and converges towards the global minimum (optimal solution). The cycle-to-cycle variability exhibited in VO₂ devices can naturally create such energy

fluctuations. Upon reaching the global minimum, the SHIL signal maintains the solution in a stable state.

We inject a sinusoidal signal, with a frequency twice that of the coupled network, through an additional capacitance ($C_{SHIL}$), as shown in Fig. 4. The introduction of SHIL also helps strengthening coupling in the network, enabling adjustment of the oscillators' frequencies to mitigate uncertainty in system response, particularly for oscillators exhibiting high device-to-device variability[28,35,43]. The values of the injected signal amplitudes and $C_{SHIL}$ are chosen to be sufficiently high, ensuring successful synchronization beyond a threshold below which a stable binarized solution cannot exist[39]. These parameter values need to be adapted for each specific problem due to the non-balanced number of connections among nodes present in most graphs, making it impossible to satisfy all the constraints imposed by the capacitors pairing the oscillators[26]. In scenarios involving large graphs with numerous connections, the system's energy landscape becomes more complex, and a careful selection of these parameters is required to prevent staying trapped in a local minimum – a phenomenon known as the 'freeze-out' effect[41]. An experimental example of this phenomenon is shown in SI. Adjusting carefully the electrical parameters for each specific problem also maximizes performance and ensures the successful binarization of the system. Failure to do so results in large cluster diameters with nodes found in not well defined phase groups, as demonstrated in SI.

The dynamics of an oscillator network implement the gradient descent of the Ising ground state problem, with the solution space being explored through the circuit's inherent physical behavior rather than in a sequential manner[42]. Therefore, the minimization of the Ising Hamiltonian in Eq. (1), directly mapped onto a system of coupled oscillators as in Fig. 2, maximizes the cut-set for any given problem[19].

Figure 5 illustrates typical results achieved for graphs involving four to nine VO₂ oscillators. The influence of the parameters $C_c$, $C_{SHIL}$, and $V_{SHIL}$ on the spread of the phases becomes evident in graphs B, D, E, F, and G (see Table S4 (SI)). In these experiments, a weaker SHIL signal amplitude or coupling results in a larger cluster diameter in both

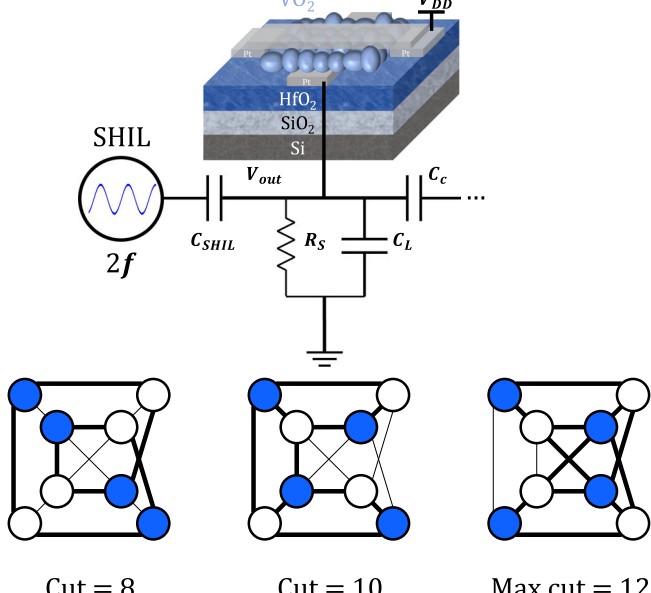

**Fig. 4 | ONN mapping of the Max-cut problem.** Injection of the SHIL signal at twice the frequency that of the coupled oscillators in the network. This technique ensures synchronization of every oscillator in one of two defined phase states. The Max-cut problem consists in finding a partition of a given graph's nodes into two sets to maximize the number of edges between the sets.

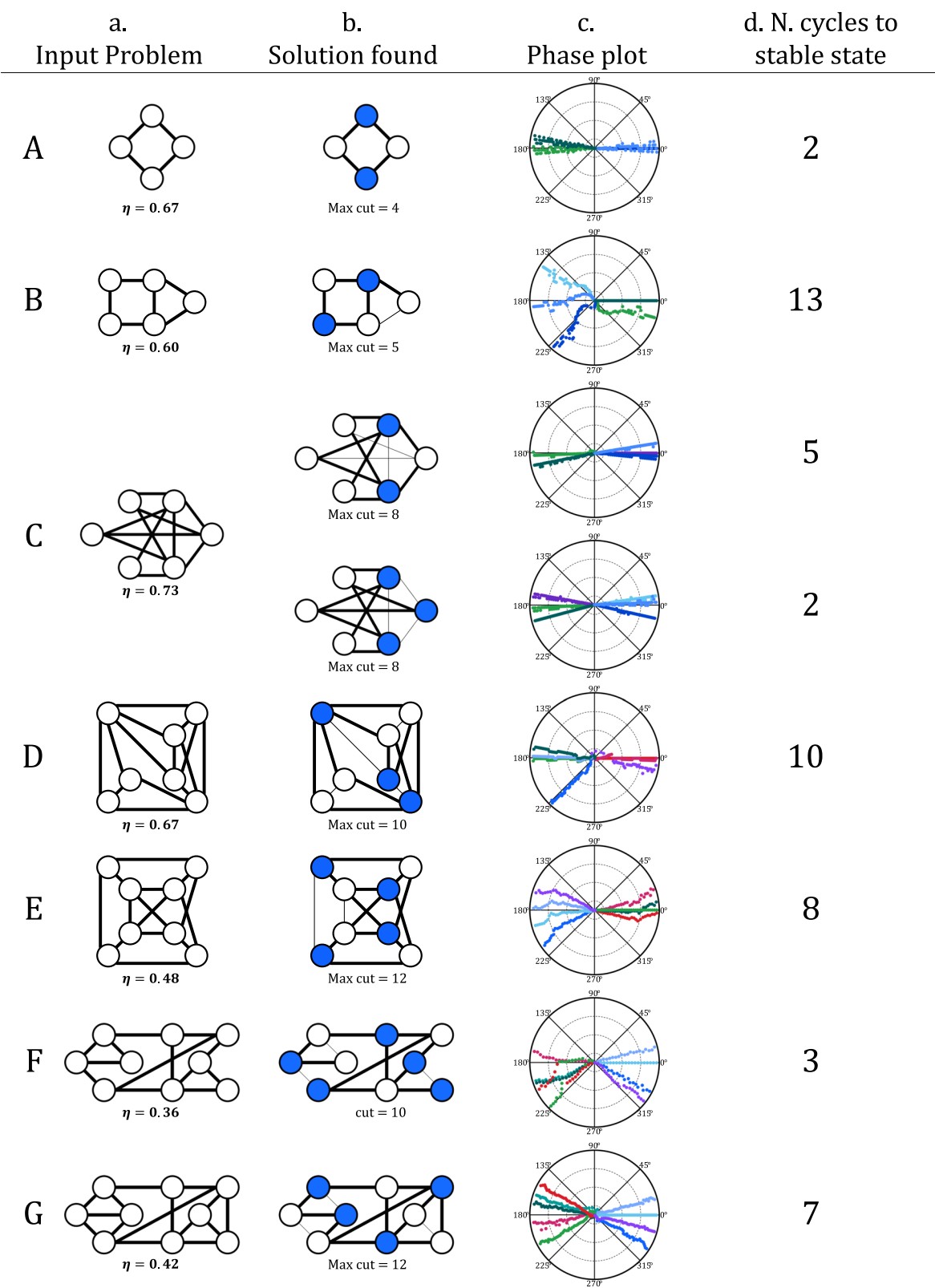

**Fig. 5 | Experimental results of coupled relaxation VO$_2$ oscillators, involving 4 to 9 nodes to solve the Max-cut problem. a** Input problem, **b** Solution found, and **c** measured Phase relationships of the ONN outputs. **d** Number of cycles required to get to the stable state. The systems converge to the solution within 13 oscillation cycles only when the connection density of the graph η > 0.4. The oscillators are partitioned into two states: one with a near 0-degree in-phase relation and the other with a close to 180-degree out-of-phase relation with respect to a reference device.

states 0 and π, as illustrated in the corresponding phase plots in Fig. 5. Another important property is the connection density η, defined as the ratio of the number of edges within the graph to the total number of edges in a same-sized fully connected graph[44]. In instances where the graph is sparse with η < 0.4, such as graph F, achieving a balance between all parameters for successful network binarization while attaining the ground state proves difficult. This observation aligns with previous research findings[44]. The issue arises when dealing with sparse graphs (η < 0.4), as their equivalent adjacency matrix has significantly different eigenvalues, referred to as a large spectral radius, making it challenging for the SHIL signal to effectively divide the graph's nodes into only two distinct groups[44]. Although SHIL is essential for discretizing the solution, it also introduces limitations on the synchronization dynamics of the system that potentially lead to suboptimal solutions[43,45].

Figure 6 presents the results of repeated Max-cut experiments on graphs C, D, E, and G, using six to nine interconnected $VO_2$ oscillators. Similar to the findings demonstrated in Fig. 5, the ONNs shown in Fig. 6 consistently reach their stable states within 15 oscillation cycles. This highlights the potential for faster time execution of massive parallel computations through coupled oscillators[46]. It should be noted that most graphs in Figs. 5 and 6 are either planar, i.e. they can be drawn on a plane without edges intersecting, or nearly planar[40]. Although algorithms exist to solve the Max-cut problem in polynomial time for planar graphs[40], our results in Figs. 5 and 6 demonstrate notable efficiency by attaining solutions within 15 oscillation cycles. It will be essential to reevaluate this level of performance when dealing with denser non-planar graphs involving a greater number of $VO_2$ oscillators.

Figure 6 shows that in the case of graph C, which has the highest connection density (η > 0.7), the network attain stability in the ground state in over 62% of the trial runs. For all graphs except D, the best solution is found in the majority of cases, exceeding 42% for the largest graph employing nine oscillators. These already promising outcomes could be further enhanced by incorporating various techniques beyond the Kuramoto model tailored for oscillator-based computing[39]. This could involve modifications to the shape of oscillations or the SHIL signal, by incorporating additional terms from the Fourier series expansion[39,41]. Such adjustments have demonstrated improved capabilities in computing the optimal solution[41].

## Max-3SAT problem

The 3-SAT problem consists in finding a Boolean combination of variables in a way that satisfies a given formula $\mathcal{F}$[3]. This formula is made up of smaller parts called clauses $\boldsymbol{C}$, and each one contains three specific pieces of information called literals[3].

$$\mathcal{F} = C_1 \wedge C_2 \wedge \ldots \wedge C_{M-1} \wedge C_M \tag{2}$$

Where $\boldsymbol{C_j}$ is the inclusive disjunction of three literals $x^j$, such that

$$\boldsymbol{C_j} = \left( x_a^j \vee x_b^j \vee x_c^j \right) \tag{3}$$

Having a dedicated hardware solver for 3-SAT presents a significant potential to accelerate the computation of NP-complete problems, as these problems can all be reduced to 3-SAT[3]. The mapping of the Ising Hamiltonian formulation in Eq. (1) to our $VO_2$ ONN hardware involves using capacitors as coupling coefficients and introducing an additional signal with the same frequency as the coupled oscillators to act as an external bias $\boldsymbol{h_i}$. The circuit equivalent of a single node and edge is illustrated in Fig. 7.

In this configuration, the system seeks the maximum number of satisfied clauses $\boldsymbol{K}$, effectively solving the NP-hard Max-3SAT problem by identifying independent sets of nodes that are not connected. To construct the equivalent graph, each literal is mapped to a node ($VO_2$ oscillator), while each clause corresponds to an interconnection (capacitor), forming the triangles depicted in Fig. 7. The goal is to ensure that at most one node per clause is part of the independent set[38]. Furthermore, all complementary literals are interconnected (blue connections in Fig. 7), as they cannot simultaneously satisfy the two corresponding clauses. The size of the independent set determines the count of satisfied clauses, and solving the Max-3SAT is equivalent to identifying the largest independent set within the corresponding graph.

Figure 8 presents the results of repeated Max-3SAT experiments on graphs involving six ($\mathcal{F}_1$) to nine ($\mathcal{F}_2$ and $\mathcal{F}_3$) oscillators. The values of the electrical circuit parameters chosen for each graph are reported in Table S5 (SI).

$$\mathcal{F}_1 = (x_1 \vee \overline{x_2} \vee x_3) \wedge (x_1 \vee x_2 \vee \overline{x_3}) \tag{4}$$

$$\mathcal{F}_2 = (x_1 \vee \overline{x_2} \vee x_3) \wedge (\overline{x_1} \vee \overline{x_2} \vee \overline{x_3}) \wedge (x_1 \vee x_2 \vee \overline{x_3}) \tag{5}$$

$$\mathcal{F}_3 = (x_1 \vee \overline{x_2} \vee x_3) \wedge (\overline{x_1} \vee x_2 \vee \overline{x_3}) \wedge (x_1 \vee \overline{x_2} \vee \overline{x_3}) \tag{6}$$

The size of the independent set $\boldsymbol{K}$ is measured solely under the condition that the oscillators which exhibit an in-phase relationship with the bias $\boldsymbol{h}$ (noted as positive spin ↑) are not interconnected. In cases where they are interconnected, the system converges to a stable state not corresponding to an independent set in our mapping. This results in an undefined count of independent sets, noted ∅. It is worth noting that in instances where nodes ($VO_2$ oscillators) represent logical complements but still oscillate in-phase with a negative spin ↓, the system fails to assign a true/false value to the associated variable. This situation is depicted in the third Max-3SAT graph of Fig. 7, featuring the variables $x_a$ and $\overline{x_a}$. In such cases, the calculated value of $\boldsymbol{K}$ is not associated with a logical/defined combination of variables within the Boolean expression.

Figure 8 shows the system's capacity to identify the Max-3SAT in all instances $\mathcal{F}_1$, $\mathcal{F}_2$, and $\mathcal{F}_3$, with a success rate reaching up to 75% in the case of six coupled oscillators. Across all experimental trials, the network consistently achieved stability within 23 oscillation cycles, a significant improvement compared to digital methods. To potentially enhance the system's ability to solve Max-3SAT, an approach could involve balancing and adapting better the parameters for each specific graph. For example, ensuring the coupling coefficients $J$ are higher

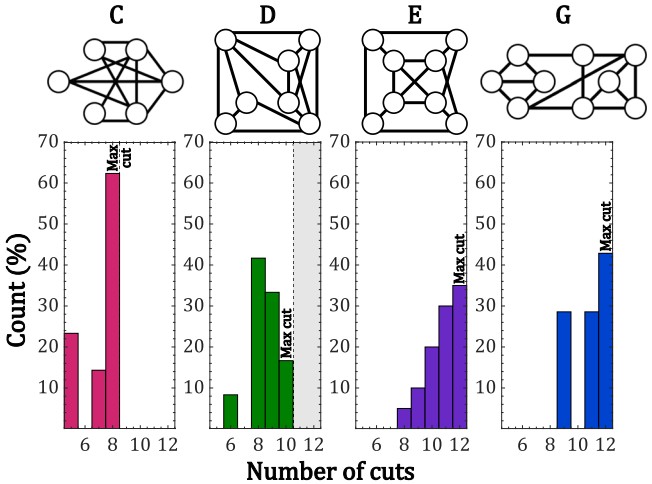

**Fig. 6 | Distribution of the cuts attained for graphs involving 6 to 9 coupled $VO_2$ oscillators to solve the Max-cut problem.** The optimal solution is found in most instances for graphs C, E, and G.

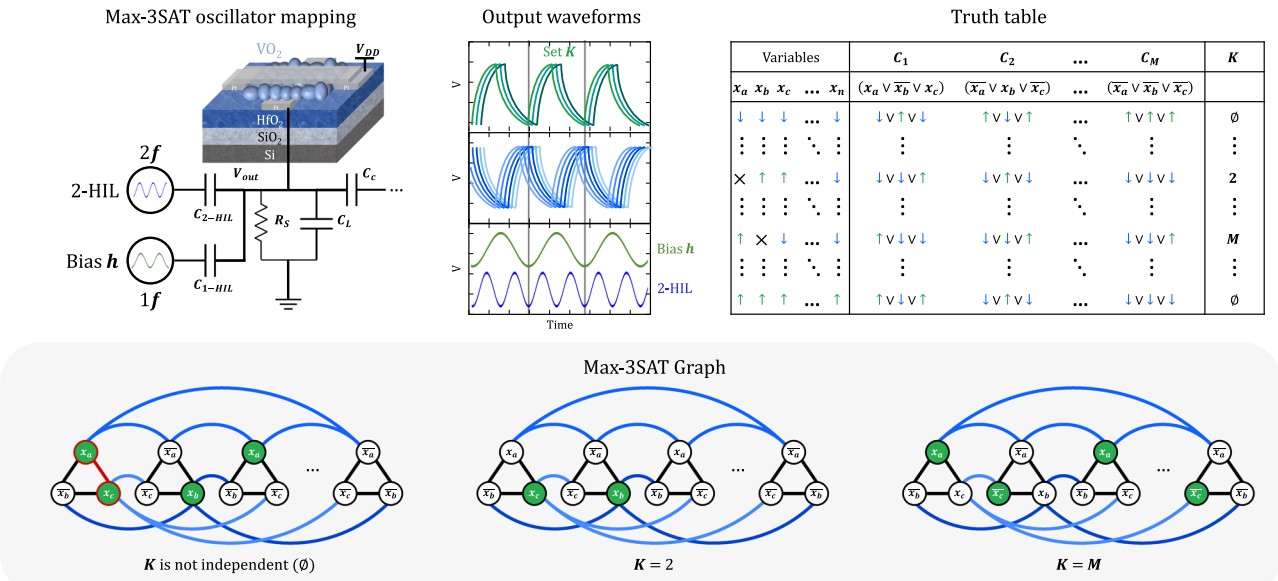

**Fig. 7 | ONN mapping of the Max-3SAT problem.** The problem is solved through the injection of two SHIL signals: one at the same frequency (bias $h$) of the coupled oscillators in the network, and the other at twice the frequency to ensure binarization. The Max-3SAT consists in finding an assignment of true and false ($\uparrow$ and $\downarrow$) values to variables that satisfies the maximum number of clauses in a Boolean formula, where each clause contains three literals. The maximum independent set $K$ is calculated under the condition that variables with a positive spin $\uparrow$ (oscillators in-phase with the bias $h$) are not interconnected. In cases where two nodes (VO$_2$ oscillators) represent logical complements but share the same spin value $\downarrow$ (both out-of-phase with the bias $h$), the system fails to assign a true/false value to the associated variable, which is denoted by .

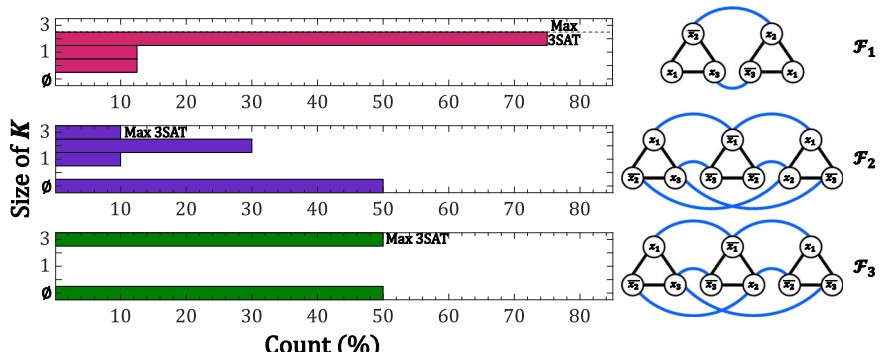

**Fig. 8 | Distribution of the independent set size $K$ attained for graphs involving 6 to 9 coupled VO$_2$ oscillators to solve the Max-3SAT problem.** The optimal solution is found in all instances, proving 3-SAT solvers can be implemented with VO$_2$ oscillators.

than the bias $h$ would prevent getting trapped in local energy minima and ensure oscillators representing complement variables stabilize with opposite spins. The chosen amplitudes ($V_h$ and $V_{2\text{-HIL}}$) and capacitances ($C_{1\text{-HIL}}$ and $C_{2\text{-HIL}}$) of the injected signals also impact significantly the eventual convergence state of the network. Further exploration of the influence of these values is required to maximize the likelihood of successfully solving the Max-3SAT problem. Nonetheless, our results presented in Fig. 8 serve as a proof of concept, showcasing a network of VO$_2$ oscillators capacitively coupled and influenced by SHIL signals can successfully realize an efficient and quick analog 3-SAT solver.

## Discussion

Table 1 provides a comparison between our work and previous studies on the coupling of oscillators to build Ising machines for optimization problem-solving. It highlights the limitations of traditional architectures relying on CMOS technology, such as excessively long computing times[7] or high energy consumption[47], despite operating at considerably higher frequencies. Unreported experimental results on

these figures, as in Ahmed et al. and Tatsumura et al.[24,48], may imply unfavorable outcomes.

In our study, we investigated VO$_2$-based ONNs at low frequencies to demonstrate experimentally their ability to solve COPs. However, the true potential of these networks lies in their scalability down to nanometric sizes, enabling ultralow power consumption[46] (around 13 μW/oscillator) and rapid convergence[49] (time to solution <1 μs) to optimal solutions with high accuracy within just a few oscillation cycles. In-depth scaling challenges analyses are reported in Delacour et al.[46] and Carapezzi et al.[49]. The inherent parallelism of ONN computing allows for short computing times, ranging from tens of nanoseconds to a few microseconds, even with the current oscillation frequencies in the MHz domain. This positions energy-efficient phase-binarized oscillators as an attractive choice to outperform CMOS-based technologies in addressing NP-complete problems, a finding also reported in Singhal et al.[25]. However, the circuit implementation of large-scale ONNs poses a challenge due to the quadratic increase of coupling elements. The on-chip coupling implementation of larger networks can be achieved by using programmable memristive arrays

**Table 1 | ONN-based Ising machine benchmark**

| Technology | Max graph size | Measured time to solution (µs) | Operating frequency (kHz) | Accuracy | Average power (µW/oscillator) |
|---|---|---|---|---|---|
| FPGA SB-based Ising machine with 8 processors[48] | 16,384 | 1200 | 262,000 | Not mentioned | Not mentioned |
| Schmitt trigger oscillators[52] | 30 | Not mentioned | ~44 | 0.72 | 59 |
| Ring oscillators[24] | 560 | Not mentioned | 118,000 | 0.82 | 41 |
| Asynchronous digital logic + analog oscillator (CMOS 180 nm)[7] | 282 | ~7.6 × 10^6 | ~0.21 | Not mentioned | Not mentioned |
| CMOS LC oscillators[47] | 240 | 3500 | 5 000 | ~0.13 | 21,000 |
| Coupled VO$_2$ oscillators (This work) | 9 | 11,500 | ~2 | ≤0.75 | 180 |
| Coupled VO$_2$ oscillators (Projected work[46,49]) | >64 | ~1 | ~30,000 | >0.9 | 13 |

or capacitive banks selected through multiplexers. Such high level of adaptability is needed to connect nodes based on the unique requirements of each optimization problem. To address scalability more efficiently, an alternative approach inspired by graph theory lemmas involves partitioning a large graph into smaller, more manageable graphs[3]. For example, the Traveling Salesman Problem could be solved by accumulating solutions from small-scale ONNs to establish the shortest path, and the same reasoning could be extended to Max-cut, Max-3SAT, and other optimizations problems. This would prevent bulky coupling matrices from occupying a substantial chip area, particularly in problems with thousands of nodes requiring as many coupled oscillators.

When these NP-complete problems are large-scale, their intrinsic complexity increases the risk of getting trapped in a local minima, resulting in suboptimal solutions. Previous studies addressed this challenge by employing various techniques to overcome energy barriers in systems with such complex energy landscapes having multiple local minima[19,26,41]. These techniques included incorporating decaying noise as the system approached its ground state or introducing SHIL signals via simulated annealing, where signal amplitude is varied to facilitate exploration of the solution space. These methods proved essential to solve NP-complete problems using oscillation-based computing[19,26,41]. However, when applied to our specific experiments focused on Max-cut and Max-3SAT problems, these techniques failed to yield any improvements. The results obtained from their implementation were inconclusive and no definitive conclusions could be drawn regarding their benefits. We argue that the intrinsic noise present in all crossbar oscillators, combined with device-to-device variability, introduces a satisfactory level of randomness, enabling the network to effectively explore the entire solution space. This observation holds true in our experimental setup with small graphs connecting up to nine oscillators, and may vary when dealing with larger problems[25].

Additionally, when selecting the electrical parameters for a specific problem, it is imperative to consider the degree of connectivity associated with each node within the graph. When the graph is unbalanced, as in graph F from Fig. 5 where certain nodes have as few as 2 connections and others as many as 4, our findings suggest that reaching the best solution becomes more complex or unattainable. On the other hand, establishing a more even distribution of connections within the graph, as seen in graph G where most nodes have 3 or 4 connections, facilitates the identification of acceptable circuit parameters, ultimately leading to successful convergence towards the optimal solution.

We have demonstrated how the dynamic behavior of coupled VO$_2$ oscillators can be harnessed to solve complex optimization problems, including Graph Coloring, Max-cut, and Max-3SAT problems. Our approach involves the development of task-specific mappings from the Ising Hamiltonian formulation to our network of relaxation oscillators, with up to nine crossbar VO$_2$ devices. Through our experimental

findings, we show that the system achieves stability in less than 25 oscillation cycles, indicating the potential for faster time execution of extensive parallel computations using large-scale networks of interconnected oscillators[25,46]. Using sub-harmonic injection locking, we binarize the solution provided by the oscillators into two distinct phase groups, corresponding to "spins up" and "spins down" in the Ising formulation of Max-cut and Max-3SAT problems. We demonstrate that graphs exhibiting high connection density ($\eta > 0.4$) when mapped onto our VO$_2$ network tend to converge more readily towards their optimal solution due to the typically smaller spectral radius of the equivalent adjacency matrix.

The proposed architecture offers a high degree of configurability, enabling the emulation of associative neural network capabilities without the requirement of complete all-to-all connectivity[50,51]. This versatility and configurability are advantageous for addressing problems that demand specialized solutions, as they exhibit varying levels of complexity and inter-unit connectivity[15,20]. The compact nature of the VO$_2$ oscillators, connected via nanoscale memristive synaptic and resistive units (positive coupling) or capacitors (negative coupling), as demonstrated in this study, highlights the ease with which practical hardware implementations can be realized to build customizable analog solvers. Their intrinsic physical characteristics embedding memory and computation on the same platform can tackle complex optimization tasks, a feat that has proven inefficient through traditional digital computing[15,39]. These oscillators constitute foundational building blocks for application-specific functions, offering the potential for seamless integration on a CMOS-compatible medium to address multifaceted technological challenges across industries.

## Data availability

The data generated in this study is available on Zenodo under the accession code https://doi.org/10.5281/zenodo.10879440.

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

## Acknowledgements
This project has received funding from the EU's Horizon program under projects No. 871501 (NeurONN), 861153 (MANIC), and 101092096 (PHASTRAC). The authors thank the Cleanroom Operations Team of the Binnig and Rohrer Nanotechnology Center (BRNC) for their help and support.

## Author contributions
O.M. and N.H. fabricated and characterized the devices. O.M. and C.D. performed data analysis. O.M. and M.J. set up the experimental measurements. M.J., J.N., and M.J.A conceptualized the implementation basis of the computing problems. S.K., M.J.A., B.L.B., G.I., and A.T.S. supervised and directed the project. O.M. wrote the manuscript. All authors commented on the manuscript.

## Funding

## Competing interests
The authors declare no competing interests.
