## [Peer Review File · Nature Communications]

REVIEWER COMMENTS

Reviewer #1 (Remarks to the Author):

The authors report on coupled VO₂ oscillators for solving optimization problems. The oscillations are due to conductance changes in this material when heated above a certain temperature. This is an active area of research concerning new materials for neuromorphic-type computing and related applications.

Technical comments for the authors to consider:

- The authors should show the temperature dependent oscillator characteristics and how the coupling and computing performance varies with the temperature. This is important practical consideration to understand the properties / use cases of their devices.

- The authors use msec timescales if I understand correctly. Is this because they want each device to return to initial state fully? What happens if the devices are at different temperatures?

- The authors argue that their devices can be used to reach sub-microseconds speeds by scaling etc. however they have not discussed the refractory period noted for VO₂ oscillators (Lin et al, Front Neurosci, 12, Art. 856, 2018) or the long-lived metastable states when VO₂ device is driven out of equilibrium by electric stimulus (Sood et al, Science, 373, 252, 2021). I think these factors are important to discuss in the projected behavior of their coupled oscillators especially since the authors are interested in hafnia underlayers for VO₂ growth.

- I did not find any details of the individual device in the main or supplement file. Please provide this information. I.e what are the electrodes? VO₂ film thickness? How are the devices connected to each other? Are these individual devices connected by macroscopic wires or entire network is fabricated in one chip?

- Please compare results to prior literature on similar topic:

Tobe, Ryuta, et al "Coupled oscillations of VO₂-based layered structures: Experiment and simulation approach." Journal of Applied Physics 127.19 (2020).

Velichko, Andrey, et al. "Thermal coupling and effect of subharmonic synchronization in a system of two VO₂ based oscillators." Solid-State Electronics 141 (2018): 40-49.

Reviewer #2 (Remarks to the Author):

Review of manuscript NCOMMS-23-48177:

“A CMOS-compatible oscillation-based VO2 Ising machine solver” by Olivier Maher et al.

Authors demonstrated an Ising model solver based on networked VO2-based oscillators integrated on a Silicon platform. The dynamics of VO2-based oscillators has a degree of freedom in phase space and the phase can be stabilized to binary states by using sub-harmonic injection locking. Spin up and down state of the Ising model was represented by these bistable states. Authors confirmed that their VO2 oscillator network can reach optimal solutions for Map coloring, Max-cut, and 3-SAT problems in various graph instances with up to 9 nodes.

Although the tested graph instances are very small-scale and the success probability is still not significant high, the presented system shows a unique property in which different optimization process in 2-dimensional state (Map coloring) and binary state (Max-cut, 3-SAT) can be changed flexibly by using external injection to the system. Authors investigated the dynamics of networked oscillators in solving optimization problems with various operating parameters, and I guess these results and discussion in this article will be of interest to readers in a position to investigate such combinatorial optimization problems. The manuscript is, in my opinion, deserves the publication in Nature Communications after authors addressed some points listed below.

(1) About connectivity of VO2 oscillators, some information should be included in this article as follows:

- Authors describe that the coupling between VO2 oscillators is implemented with a capacitance, and the phases of coupled oscillators tend to stabilize in the out-of-phase configuration. It means that parameters in the adjacency matrix is limited to negative value and only anti-ferromagnetic coupling in Ising model can be implemented. For solving Ising problem in universal network structures, ferromagnetic coupling is also necessary for the system. Authors should mention about the way or feasibility of implementation of the ferromagnetic couplings to the presented system.
- The coupling strength is defined by the capacitance C_c of each connection. Is it possible to programmable control the capacitance value for changing network structures after fabrication of the device?
- For scaling-up the presented system, authors should mention their strategy to implement large-scale network based on such coupling with capacitance. For example, about 200,000 capacitance couplings can be required for solving 1000-node Ising problem with network density 0.4.

(2) In this article, authors describe that their VO2 oscillator emulate neuron (line 111) and use words “synaptic units” for their couplings (line 471). For general readers of this journal, please add explanation what kind analogy is existing between the VO2 network and biological neural networks.

Reviewer #3 (Remarks to the Author):

The authors present a VO2 based oscillator Ising machine showcasing the two different operating modes, namely with and without external harmonic injection. The motivation for using VO2 for realizing the oscillators is the compact nature of the resulting oscillator design. The resulting characteristics of the coupled VO2 oscillator system are utilized for solving the graph coloring problem (without injection) and MaxCut / Max-3-SAT problem (with injection), respectively.

While individual dynamics (with and without injection, as referenced by the authors) have been presented in individual works earlier, this work aims brings them together, backed by experiment. This can help showcase the computational characteristics of the coupled oscillator platform as a whole, making it potentially interesting.

However, I recommend that authors address the following aspects:

1. Graph coloring: It seems from Fig. 3 (graph in the last row) that the different clusters representing the colors are not evenly separated. It would be useful if the authors can explain this. Further, how does the asymmetry in the degree of each graph node (regularity) impact the cluster diameter / uniformity of spacing? I note that graph 3 is the only graph which is non-planar and where the nodes do not have the same degree.
2. I observe that many of the graphs considered in Fig. 2 are planar or almost-planar graphs. Planar graphs are known to be solvable in polynomial time for graph coloring and MaxCut. How does the system perform when solving non-planar graphs? In my opinion, this assessment will be important for analyzing the computational properties of the system.
3. From table 1, the values of the coupling capacitors are in the nF (nano Farad) range. It seems like implementing such large capacitors at practical scales would be impractical, considering that the coupling network would consume a significant fraction of the hardware. I suggest that the authors clarify how these parameters would scale (and why) when using smaller devices?
4. I recommend that the authors address and benchmark the question of scalability of the system with some quantitative metrics (energy-to-solution, success probability etc.)

5. Suggested clarification: The authors motivate their work as a pathway to accelerating NP problems. While this is valid and true, I recommend that the authors clarify that their approach would belong to the heuristic domain (particularly considering that they non-exponentially increasing run-time).

Reviewer #1

Point 1:

The authors should show the temperature dependent oscillator characteristics and how the coupling and computing performance varies with the temperature. This is important practical consideration to understand the properties / use cases of their devices.

The operation temperature is now clearly stated in Figure 1's caption:

d) Oscillation measurements of a single VO₂ crossbar oscillator at 220 K (active area: 80 × 80 × 60 nm³, R_s = 50 kΩ, C_L = 11.27 nF, V_{DD} = 5 V).

And the reader is pointed to Supplementary Information for more information on temperature dependent characteristics:

The typical oscillating behavior of one device is shown in Figure 1d. Low temperature operation is further motivated in SI.

The following sections have been added to Supplementary Information to answer questions raised by the reviewer that the reader might have:

Figure S2. a) Uncoupled and b) coupled oscillating characteristics of two VO₂-based oscillators. The oscillation amplitude ($V = V_{DD} - V_{out}$) measured at the device terminals decreases with increasing temperature (active area: $100 \times 50 \times 60 \text{ nm}^3$).

Table S1. Values of the circuit parameters employed to conduct the temperature dependent measurements in Figure S2.

Temperature (K)	V_{DD} (V)	R_s (k Ω)	C_L (nF)	C_c (nF)	V_{TL} (V)	V_{TH} (V)	$f_{natural}$ (kHz)	$f_{coupled}$ (kHz)	N. cycles to stable state when coupled
220	5.50	40	10.0	0.68	0.46	2.60	2.90	2.75	5
240	5.50	40	10.0	0.68	0.45	2.15	3.35	3.20	5
260	5.75	40	10.0	0.68	0.40	1.80	4.05	3.90	7
280	5.85	40	10.0	0.68	0.34	1.44	4.40	4.07	3
300	4.50	30	10.0	0.68	0.29	1.12	4.90	4.75	5
320	7.00	30	10.0	—	—	—	—	—	—

Figure S2 presents the oscillating characteristics of two uncoupled VO₂-based oscillators in a) and coupled in b). The oscillation amplitude ($V = V_{DD} - V_{out}$) measured at the device terminals decreases progressively with increasing temperature. This reduction is attributed to the lower power required to reach the material's switching points (see Figure S3d) when the ambient temperature is higher. This trend is further illustrated by the reducing threshold voltages (V_{TL} and V_{TH}) reported in Table S1. The number of cycles required to reach the expected out-of-phase relationship, when the devices are coupled, remains fairly stable regardless of the ambient temperature.

Figure S3. a) Output signals of two VO₂ devices connected to a series resistance ($R_s = 10 \text{ k}\Omega$) biased with a voltage ramp ranging from 0 V to 6 V. b) Experimental waveforms of two uncoupled VO₂ oscillators at 320 K (active area: $100 \times 50 \times 60 \text{ nm}^3$, $R_s = 5 \text{ k}\Omega$, $C_L = 10.0 \text{ nF}$, $V_{DD} = 875 \text{ mV}$). c) I-V (current-driven) graph of a VO₂ device at 250 K. (active area: $100 \times 50 \times 60 \text{ nm}^3$). d) Resistance-temperature (R-T) characteristics of a VO₂ device (active area: $100 \times 50 \times 60 \text{ nm}^3$, $R_s = 40 \text{ k}\Omega$).

In Figure S3a, the voltage at the devices' outputs at 220 K, connected to a low series resistance ($10 \text{ k}\Omega$), are shown as the supply voltage is ramped. Oscillation occurs only within a specific voltage range (V_{osc}). With an increase in ambient temperature, this range becomes smaller, and the variability among VO₂ devices³³ makes it challenging to find a supply voltage that aligns within V_{osc} for all devices simultaneously. This situation is represented in Figure S3b, where, at 320 K with the same V_{DD} , one device remains in a high (insulator) resistive state while the other remains in a low (metallic) resistive state. In this case, the absence of an oscillating pattern prevents successful device coupling. This motivates our decision to operate at low temperature (220 K) to ensure a broader range of viable operation modes under large biases (see Figure S3c).

Point 2:

The authors use msec timescales if I understand correctly. Is this because they want each device to return to initial state fully? What happens if the devices are at different temperatures?

The timescale is determined by selecting a load capacitance ($C_L \geq 10 \text{ nF}$) across all experimental setups, which slows down oscillations to the millisecond range. The use of such high capacitance values is beneficial for achieving improved signal resolution, especially when measuring output signals with our experimental setup's limited sampling frequency. Parasitic capacitances introduced by cabling from coupling units and the measuring system become negligible compared to C_L , and hence have nearly no influence on the oscillation behaviors of our devices. This is why our study is conducted

in the millisecond range, and it is not dependent on waiting for each device to return to its initial state. The devices consistently oscillate between the same two defined states, as illustrated in Figure 1d. This motivation is explicitly stated in the text just before Figure 1, with **now a reference** to the first source this reviewer mentioned in Point 5, which provides an identical explanation:

The addition of an external load capacitor connected off-chip, as shown in Figure 2, is employed to achieve uniform frequency operation among the oscillators and compensate for variations in parasitic capacitances resulting from individual contacts to our VO₂ devices (see Figure 1c).³¹

Questions about temperature were answered in Point 1.

Point 3:

The authors argue that their devices can be used to reach sub-microseconds speeds by scaling etc. however they have not discussed the refractory period noted for VO₂ oscillators (Lin et al, Front Neurosci, 12, Art. 856, 2018) or the long-lived metastable states when VO₂ device is driven out of equilibrium by electric stimulus (Sood et al, Science, 373, 252, 2021). I think these factors are important to discuss in the projected behavior of their coupled oscillators especially since the authors are interested in hafnia underlayers for VO₂ growth.

Our VO₂ devices are combined with a series resistance (R_s) and a load capacitor (C_L) to create relaxation oscillators, which have a different operation mode from the spiking VO₂ unit presented in Lin et al, Front Neurosci, 12, Art. 856, 2018 – see Figure 1a and corresponding section. This oscillating operation mode does not lead to a synaptic refractory period since C_L discharges through R_s only, even when operated at higher frequency, as simulated in ref⁵⁰ (S Carapezzi *et al*, How Fast Can Vanadium Dioxide Neuron-Mimicking Devices Oscillate? Physical Mechanisms Limiting the Frequency of Vanadium Dioxide Oscillators 2023 *Neuromorph. Comput. Eng.* **3** 034010). This has now been emphasized when describing the operation of the VO₂ oscillators in Methods, referencing the publication from Lin et al. (ref³²) and results we obtained in ref³³:

The addition of an external load capacitor connected off-chip, as shown in Figure 2, is employed to achieve uniform frequency operation among the oscillators and compensate for variations in parasitic capacitances resulting from individual contacts to our VO₂ devices (see Figure 1c).³¹ This configuration prevents refractory period effects observed in other studies³² and results in stable oscillations, with less than 3.5% amplitude variation from cycle to cycle (see Figure 1d).³³

In this paper, we focus on the potential applications of coupled VO₂ networks to solve combinatorial optimization problems. The metastable states of VO₂ driven electrically as well as the impact of HfO₂ and several other materials are studied in details in another manuscript (ref33 – Maher *et al*, Highly Reproducible and CMOS-Compatible VO₂-Based Oscillators for Brain-Inspired Computing, Scientific Reports (under review)) that focuses on material integration and related coupling challenges. While we recognize the

importance of discussing these factors to understand the material limitations associated with VO₂ coupled devices, such a discussion is more appropriately covered in our other manuscript. We feel that including it here may impact the readability of the paper.

Point 4:

I did not find any details of the individual device in the main or supplement file. Please provide this information. What are the electrodes? VO₂ film thickness? How are the devices connected to each other? Are these individual devices connected by macroscopic wires or entire network is fabricated in one chip?

The detailed fabrication of individual devices is provided in ref32 – Maher *et al*, Highly Reproducible and CMOS-Compatible VO₂-Based Oscillators for Brain-Inspired Computing, Scientific Reports (under review). We agree that a summary of this description should be available to the reader of this manuscript, and have now included the following in the Supplementary Information, which is now clearly referenced in the main text:

Main text: The detailed fabrication process and the basics on fundamental operation can be found in ref³² and in Supplementary Information (SI).

In SI:

Figure S1. Fabrication process of VO₂ crossbar devices.

Figure S1 illustrates the fabrication process of the VO₂ crossbar devices. A thin layer (10 nm) of hafnium oxide (HfO₂) is deposited by atomic layer deposition (ALD) on a Silicon (Si) wafer (500 nm) with native silicon dioxide (SiO₂) (~2 nm). The bottom Platinum (Pt) electrodes are defined through e-beam lithography and deposited by metal evaporation (30 nm). Following the lift-off process, the amorphous vanadium oxide film is deposited by ALD (60 nm) with the Tetrakis[ethylmethylamino] vanadium (TEMAV) reaction at 150

°C, using Argon as inert carrier gas and water as the oxidation agent. The wafer is annealed at 520 °C for 10 minutes under a constant oxygen partial pressure of 50 μ bar to stabilize the film in the VO₂ state – see ref³³ for more details. A lithography process followed by a sulfur hexafluoride (SF₆) plasma-based dry etch defines the device area. The top Pt electrodes (95 nm) are subsequently deposited using the same method as the bottom electrodes.

Clarification on how the graphs were physically realized have now been added as follows to show that the entire networks were not fabricated on a single chip yet:

The introduction of this **external** series resistance generates a dynamic voltage divider, establishing the relaxation oscillation nature of the system with fixed limit cycle, amplitude, and frequency. The addition of an external load capacitor **connected off-chip**, as shown in Figure 2, is employed to achieve uniform frequency operation among the oscillators and compensate for variations in parasitic capacitances resulting from individual contacts to our VO₂ devices (see Figure 1c).

In the case of electrical oscillators, such as our VO₂-based relaxation oscillators, the **off-chip implementation** of passive and capacitive interconnection provides direct and sufficiently strong interaction to bring the oscillators into frequency locking.

Point 5:

Please compare results to prior literature on similar topic: Tobe, Ryuta, et al "Coupled oscillations of VO₂-based layered structures: Experiment and simulation approach." Journal of Applied Physics 127.19 (2020). Velichko, Andrey, et al. "Thermal coupling and effect of subharmonic synchronization in a system of two VO₂ based oscillators." Solid-State Electronics 141 (2018): 40-49

The first paper by Tobe et al presents two VO₂ oscillators fabricated on an indium-tin-oxide glass substrate coupled with a capacitor. Most of the results they present revolve around defining parameters, especially the supply voltage, to achieve in-phase or out-of-phase coupling through simulations. Since no computing is achieved in this publication, it is difficult to compare their results to ours, targeting computation applications. However, the nature of the coupling and the focus on VO₂ compare well with the fundamentals computing units we use in our manuscript. Their work was explicitly cited in the appropriate sections (ref³¹):

The addition of an external load capacitor connected off-chip, as shown in Figure 2, is employed to achieve uniform frequency operation among the oscillators and compensate for variations in parasitic capacitances resulting from individual contacts to our VO₂ devices (see Figure 1c).³¹

The VO₂ devices are fabricated on silicon platform with a hafnium oxide interlayer to create a granular film comprised between two metallic electrodes whose cross-section define an area where current can flow and generate thermal filaments through Joule heating (see Figure 1c).^{31,32}

In the case of electrical oscillators, such as our VO₂-based relaxation oscillators, the off-chip implementation of passive and capacitive interconnection provides direct and sufficiently strong interaction to bring the oscillators into frequency locking. 2,31,34

The second paper by Velichko also explores VO₂ coupling, but their synchronization mechanisms, relying on thermal coupling, vastly differ from our electronic component-dependent approach. Their study also addresses harmonics synchronization, but unlike our study, external injection signals are not used to enforce defined binarized states/spins in the oscillators. Since their paper is not centered around computation or solving optimization problems, making comparisons with our current study proves difficult. The considerations presented in their publication could become important if our devices were closely packaged, potentially influencing each other through thermal interactions.

Reviewer #2

Point 1:

About connectivity of VO₂ oscillators, some information should be included in this article as follows:

- Authors describe that the coupling between VO₂ oscillators is implemented with a capacitance, and the phases of coupled oscillators tend to stabilize in the out-of-phase configuration. It means that parameters in the adjacency matrix is limited to negative value and only anti-ferromagnetic coupling in Ising model can be implemented. For solving Ising problem in universal network structures, ferromagnetic coupling is also necessary for the system. Authors should mention about the way or feasibility of implementation of the ferromagnetic couplings to the presented system.

It is true that in this study, only anti-ferromagnetic coupling is used to perform the computing required to solve the optimization problems. This choice is motivated in the Methods section. A precision on how to implement ferromagnetic coupling has been added when describing the mapping between the Hamiltonian formulation of the problem and the ONN:

$$H = - \sum_{1 \leq i \leq j \leq n} J_{ij} s_i s_j - \sum_{i=1}^n h_i s_i \quad (1)$$

Where J_{ij} is a coupling coefficient between units i and j , which can be positive or negative and is usually achieved in an ONN using resistors¹⁰ or capacitors²⁶, respectively.

With references to publications that study other applications of VO₂ oscillators coupled with scaled positive synaptic weights.

The aforementioned explanation was also reinforced in the conclusion:

The compact nature of the VO₂ oscillators, connected via nanoscale memristive synaptic and resistive units (positive coupling) or capacitors (negative coupling), as demonstrated in this study, highlights the ease with which practical hardware implementations can be realized to build customizable analog solvers.

- The coupling strength is defined by the capacitance C_c of each connection. Is it possible to programmable control the capacitance value for changing network structures after fabrication of the device?

In this study, the coupling capacitance C_c was adjusted externally for each problem using physical electronic components to serve as a proof of concept. For large-scale problems, a programmable coupling matrix would indeed be required to offer the level of adaptability required to connect every node in the system (e.g. capacitive banks selected through multiplexers in an FPGA implementation). Such adaptive coupling scheme that can use memristive matrices or RRAM arrays is one of the focus of our EU project (PHASTRAC.eu) and will be addressed in future work. The following sentences were added to state how one might go about designing a coupling matrix for large-scale problems:

However, the circuit implementation of large-scale ONNs poses a challenge due to the quadratic increase of coupling elements. This on-chip coupling implementation can be achieved by using programmable memristive arrays or capacitive banks selected through multiplexers. Such high level of adaptability is needed to connect nodes based on the unique requirements of each optimization problem. To address scalability more efficiently, an alternative approach inspired by graph theory lemmas involves partitioning a large graph into more manageable smaller graphs.³

- For scaling-up the presented system, authors should mention their strategy to implement large-scale network based on such coupling with capacitance. For example, about 200,000 capacitance couplings can be required for solving 1000-node Ising problem with network density 0.4.

The reviewer is right in saying that the circuit implementation of large-scale ONN networks poses a challenge due to the quadratic increase of coupling elements. To scale up our system, our approach consists in applying the same assumptions used in graph theory, indicating that an hypergraph can be accurately solved by solving its smaller graphs. This works well in associative memory applications where one can show that an 8x8 image can be retrieved properly through four small 4x4 ONNs (top left, top right, bottom left, and bottom right of the large image). For optimization problems, this is highly dependent on the problem to solve. Our team is currently working on proving the Traveling Salesman Problem could be solved by accumulating the solution of smaller graphs to define the shortest path, and a similar reasoning could be applied for the Max-cut Problem. In a real circuit implementation, this would translate into several small ONNs being used to solve a large-scale problem. This effectively prevents a bulky coupling matrix of capacitors from occupying a large chip area with its complex connection scheme. To test if this could be realized with 1000 nodes, we will first work in the digital domain with more robust design against jitter noise and mismatches, using a digital capacitive bank with multiplexers to select the paths between coupled nodes.

A paragraph synthesizing the aforementioned ideas has been added to the Discussion:

However, the circuit implementation of large-scale ONNs poses a challenge due to the quadratic increase of coupling elements. The on-chip coupling implementation of larger networks can be achieved by using programmable memristive arrays or capacitive banks selected through multiplexers. Such high level of adaptability is needed to connect nodes based on the unique requirements of each optimization problem. To address scalability more efficiently, an alternative approach inspired by graph theory lemmas involves partitioning a large graph into smaller, more manageable graphs.³ For example, the Traveling Salesman Problem could be solved by accumulating solutions from small-scale ONNs to establish the shortest path, and the same reasoning could be extended to Max-cut, Max-3SAT, and other optimizations problems. This would prevent bulky coupling matrices from occupying a substantial chip area, particularly in problems with thousands of nodes requiring as many coupled oscillators.

Point 2:

In this article, authors describe that their VO₂ oscillator emulate neuron (line 111) and use words “synaptic units” for their couplings (line 471). For general readers of this journal, please add explanation what kind analogy is existing between the VO₂ network and biological neural networks.

The following sentence was added in the section describing the dynamics of coupled oscillators to link VO₂-based ONNs to biological neurons:

A VO₂-based ONN consists of a system of oscillators acting as neurons, interconnected with synaptic weights, representing the coupling strength and the memory of the network.⁹

Reviewer #3

Point 1:

Graph coloring: It seems from Fig. 3 (graph in the last row) that the different clusters representing the colors are not evenly separated. It would be useful if the authors can explain this. Further, how does the asymmetry in the degree of each graph node (regularity) impact the cluster diameter / uniformity of spacing? I note that graph 3 is the only graph which is non-planar and where the nodes do not have the same degree.

The following explanation has been added to provide more insight into the results shown in Figure 3:

In our case, the careful choice of these values resulted in a cluster diameter, which represents the maximum phase difference among oscillators sharing the same color grouping, that is relatively small.¹⁷ For the Central European and South American graphs, the cluster diameter averaged 33.5° and 30.0°, respectively. The inherent sparsity of these graphs without all-to-all connectivity makes coloring more challenging¹⁷, resulting in larger cluster diameters compared to the Northern Europe and East Asia graphs. In the South America graph, characterized by nonuniform connectivity with varying degrees of connections on each node, the combined and unbalanced repelling effect of the coupling capacitances establishes a phase ordering among the oscillators.¹⁷ This phase ordering can only approximate the minimum vertex coloring, causing uneven cluster spacing in the solution.^{17,36}

All of the graphs presented in Figure 3 are planar, since a 2D equivalent map exists for each one:

Point 2:

I observe that many of the graphs considered in Fig. 2 are planar or almost-planar graphs. Planar graphs are known to be solvable in polynomial time for graph coloring and MaxCut. How does the system perform when solving non-planar graphs? In my opinion, this assessment will be important for analyzing the computational properties of the system.

We agree that planar graphs can be solved in polynomial time for a few optimization problems. The experimental setup we used in this study is limited to contact a maximum of 9 oscillators simultaneously. More densely connected graphs (typically non-planar) have energy landscapes (see Figure 2) with several optimal solutions. In cases with a large parameter η (commonly in dense, non-planar graphs), it becomes challenging to discern whether the network made of merely 9 oscillators converges to the optimal solution due to the correct circuit implementation and parameter choices, or simply because there are too many correct solutions, and the stable state luckily landed in one of them. To distinguish unsuccessful trials from successful ones clearly, we decided to work

with graphs (see Figure 5) that have a very small number of solutions relative to the total number of nodes ($\leq \frac{1}{2^{\#\text{nodes}-1}}$). This was done so that we could extract meaningful statistics (see Figure 6) on the success rate of our implementations. With our current results serving as a proof of concept, the performance measurements of a VO₂-based system with a larger number of nodes will indeed constitute an important consideration in analyzing the computational potential of our technology in our next work.

The following was added to the Max-cut section to specify this point:

It should be noted that most graphs in Figure 5 and 6 are either planar, i.e. they can be drawn on a plane without edges intersecting, or nearly planar.³⁹ Although algorithms exist to solve the Max-cut problem in polynomial time for planar graphs³⁹, our results in Figure 5 and 6 demonstrate notable efficiency by attaining solutions within 15 oscillation cycles. It will be essential to reevaluate this level of performance when dealing with denser non-planar graphs involving a greater number of VO₂ oscillators.

Point 3:

From table 1, the values of the coupling capacitors are in the nF (nano Farad) range. It seems like implementing such large capacitors at practical scales would be impractical, considering that the coupling network would consume a significant fraction of the hardware. I suggest that the authors clarify how these parameters would scale (and why) when using smaller devices?

This is a valid question also partly raised by Reviewer #2, Point 1. Please review their initial question below, followed by our response and the respective modifications incorporated into the manuscript:

Reviewer #2, point 1: For scaling-up the presented system, authors should mention their strategy to implement large-scale network based on such coupling with capacitance. For example, about 200,000 capacitance couplings can be required for solving 1000-node Ising problem with network density 0.4.

The reviewer is right in saying that the circuit implementation of large-scale ONN networks poses a challenge due to the quadratic increase of coupling elements. To scale up our system, our approach consists in applying the same assumptions used in graph theory, indicating that an hypergraph can be accurately solved by solving its smaller graphs. This works well in associative memory applications where one can show that an 8x8 image can be retrieved properly through four small 4x4 ONNs (top left, top right, bottom left, and bottom right of the large image). For optimization problems, this is highly dependent on the problem to solve. Our team is currently working on proving the Traveling Salesman Problem could be solved by accumulating the solution of smaller graphs to define the shortest path, and a similar reasoning could be applied for the Max-cut Problem. In a real circuit implementation, this would translate into several small ONNs being used to solve a large-scale problem. This effectively prevents a bulky coupling matrix of capacitors from occupying a large chip area with its complex connection scheme. To test if this could be realized with 1000 nodes, we will first work in the digital

domain with more robust design against jitter noise and mismatches, using a digital capacitive bank with multiplexers to select the paths between coupled nodes.

A paragraph synthesizing the aforementioned ideas has been added to the Discussion:

However, the circuit implementation of large-scale ONNs poses a challenge due to the quadratic increase of coupling elements. The on-chip coupling implementation of larger networks can be achieved by using programmable memristive arrays or capacitive banks selected through multiplexers. Such high level of adaptability is needed to connect nodes based on the unique requirements of each optimization problem. To address scalability more efficiently, an alternative approach inspired by graph theory lemmas involves partitioning a large graph into smaller, more manageable graphs.³ For example, the Traveling Salesman Problem could be solved by accumulating solutions from small-scale ONNs to establish the shortest path, and the same reasoning could be extended to Max-cut, Max-3SAT, and other optimizations problems. This would prevent bulky coupling matrices from occupying a substantial chip area, particularly in problems with thousands of nodes requiring as many coupled oscillators.

Additionally, in the Discussion, we presented a benchmark table showing the anticipated performance of our network (see last row of Table 4) upon scaling the devices with capacitors well below the nF range to operate in the MHz regime. Scaling challenges are addressed in separate publications (ref⁴⁶ and ref⁵⁰), and we prefer to direct the reader to these more detailed studies, as follows, rather than lengthening our current manuscript with redundant information:

In our study, we investigated VO₂-based ONNs at low frequencies to demonstrate experimentally their ability to solve COPs. However, the true potential of these networks lies in their scalability down to nanometric sizes, enabling ultralow power consumption⁴⁶ (around 13 μ W/oscillator) and rapid convergence⁵⁰ (time to solution < 1 μ s) to optimal solutions with high accuracy within just a few oscillation cycles. In-depth scaling challenges analyses are reported in ref⁴⁶ and ref⁵⁰.

Point 4:

I recommend that the authors address and benchmark the question of scalability of the system with some quantitative metrics (energy-to-solution, success probability etc.)

The benchmark provided in the discussion presents the results of our projected scaled devices, already published by one of the co-authors – see last row of Table 4, ref⁴⁶, and ref⁵⁰. The time-to-solution and average power per oscillator are reported in this table, from which one can derive the energy-to-solution. The success probability of graphs with up to 9 nodes is reported experimentally in this manuscript as a proof of concept (see Figure 6 and 8). Expanding these results to larger graphs with hundreds of nodes in hardware supported by simulations is the focus of our current work, and will be addressed in details in our next submission.

Point 5:

Suggested clarification: The authors motivate their work as a pathway to accelerating NP problems. While this is valid and true, I recommend that the authors clarify that their approach would belong to the heuristic domain (particularly considering that they non-exponentially increasing run-time).

We agree that this should have been more explicitly mentioned in the main text and have now made the following changes:

VO₂-based oscillating neural networks are dynamical systems both complex enough to encode computationally heavy problems **within the heuristic domain** and simple enough to realize with simple connections ensuring stable problem-solving and solution convergence under small programming biases.

When oscillators are interconnected with several others like in Figure 2, the resulting phase of each oscillator is the combined effect of the repelling forces originating from each connection. One can make the most out of this property to effectively map and solve fundamental optimization problems, **prioritizing efficiency over optimality**.

The intrinsic physical phase evolution of the coupled oscillators towards this attractor state, as shown in Figure 2, is exploited to solve the Hamiltonian equation **empirically** as a complete and reliable Ising machines

REVIEWERS' COMMENTS

Reviewer #1 (Remarks to the Author):

I appreciate the authors response to the technical comments, I believe they are satisfactory and do not have further questions on the manuscript.

Reviewer #2 (Remarks to the Author):

The authors have given satisfactory answers to the questions raised, which improved the understandability and readability of the manuscript.

Reviewer #3 (Remarks to the Author):

The authors have addressed my concerns. I would recommend the manuscript for publication.